# Pterostilbene-Mediated Inhibition of Cell Proliferation and Cell Death Induction in Amelanotic and Melanotic Melanoma

**DOI:** 10.3390/ijms24021115

**Published:** 2023-01-06

**Authors:** Joanna Wawszczyk, Katarzyna Jesse, Małgorzata Kapral

**Affiliations:** 1Department of Biochemistry, Faculty of Pharmaceutical Sciences in Sosnowiec, Medical University of Silesia in Katowice, Jedności 8, 41-200 Sosnowiec, Poland; 2Silesian Park of Medical Technology Kardio-Med Silesia, M. Curie-Skłodowskiej 10C, 41-800 Zabrze, Poland

**Keywords:** pterostilbene, melanoma, anticancer activity, apoptosis, proliferation

## Abstract

Melanoma is one of the fastest-growing cancers worldwide. Treatment of advanced melanoma is very difficult; therefore, there is growing interest in the identification of new therapeutic agents. Pterostilbene is a natural stilbene that has been found to have several pharmacological activities. The aim of this study was to evaluate the influence of pterostilbene on the proliferation and apoptosis of human melanoma cells. Proliferation of pterostilbene-treated amelanotic (C32) and melanotic (A2058) melanoma cells was determined by BRDU assay. Flow cytometric analyses were used to determine cell cycle progression, and further molecular investigations were performed using real-time RT-qPCR. The expression of the p21 protein and the DNA fragmentation assay were determined by the ELISA method. The results revealed that pterostilbene reduced the proliferation of both amelanotic and melanotic melanoma cells. Pterostilbene induced apoptosis in amelanotic C32 melanoma cells, and this effect was mediated by an increase in the expression of the *BAX*, *CASP9*, and *CASP9* genes; induction of caspase 3 activity; and DNA degradation. Pterostilbene did not affect the activation of apoptosis in the A2058 cell line. It may be concluded that pterostilbene has anticancer potential against human melanoma cells; however, more studies are still needed to fully elucidate the effects of pterostilbene on amelanotic and melanotic melanoma cells.

## 1. Introduction

Malignant melanoma originating from melanocytes is one of the fastest-growing and extremely heterogeneous cancers. The occurrence of this cancer continues to increase in most white populations around the world [1]. The genetic and biochemical heterogeneity of this aggressive cancer with high metastatic potential and limited response to chemotherapeutic agents is associated, among other occurrences, with high frequency of mutations such as those in the *BRAF*, *NRAS*, and *C-KIT* genes [2]. Among all melanomas, about 2–8% are recognized as amelanotic melanoma [3]. The amelanotic subtype is a rare form of melanoma lacking melanin that often results in a delayed diagnosis. Late diagnosis, delayed treatment, and more aggressive pathological characteristics contribute to a higher risk of death and recurrence [4,5]. The molecular heterogeneity of melanoma is an obstacle to accurate diagnosis and effective treatment. Malignant melanoma cells have developed many molecular mechanisms leading to protection of abnormal melanocytes against death, including the activation of molecular pathways involved in the regulation of melanoma cell survival, as well as accumulation of gene mutations that promote the survival of altered melanocytes, which consequently lead to tumor promotion [6].

One of the main therapeutic targets for the treatment of melanoma is the induction of cell death, e.g., apoptosis [7]. Melanoma cells avoid programmed cell death through the dysregulation of the balance between proapoptotic and antiapoptotic proteins, as well as caspase activity. One of the most promising strategies is the development of new drugs or therapies targeting proteins of the BCL family, caspases, or other mediators involved in apoptosis [8]. Numerous efforts are being put toward the development of cancer therapeutic strategies with the use of plant-derived compounds [9,10]. Natural compounds, particularly polyphenols, have been proposed to be more potent anticancer drugs than synthetic drugs due to their lower adverse effects, high precision, and secure mode of action. Numerous studies have reported that polyphenols could be considered as a possible therapeutic option in the treatment of cancer cells in the future [11,12,13].

Pterostilbene (trans-3,5-dimethoxy-4-hydroxystilbene, PTB) (Figure 1a) is a naturally occurring polyphenol that has been revealed to exhibit a variety of pharmacological properties, including the desired antioxidant, anti-inflammatory, and anticancer activity [14]. It has been shown to inhibit the cell cycle and induce both apoptotic and nonapoptotic cell death in several types of cancer cells, such as breast [15], liver [16], or lung cancer [17].

The broad spectrum of biological activity of pterostilbene suggests that this compound may have a preventive and therapeutic effect on malignant melanoma. To date, there are only a few reports on the biological activity of pterostilbene in melanoma cells and the mechanism by which pterostilbene can exert potential anticancer effects on skin cancers [18]. Most anticancer drugs exert their biological activity by inhibition of the cell cycle and induction of programmed cell death, such as apoptosis, autophagy, and necroptosis [7,8]. Therefore, the objective of this study was to evaluate the influence of pterostilbene in a wide range of concentrations on the proliferation and apoptosis of human amelanotic and melanotic melanoma cells in vitro.

## 2. Results

### 2.1. Antiproliferation Activity of Pterostilbene on C32 and A2058 Melanoma Cells

The design of the experiments focused first on investigating the response of pterostilbene-treated melanoma cells. The 5-bromo-2′-deoxyuridine (BrdU) incorporation assay was used to monitor cellular proliferative activity in response to PTB treatment of melanoma cells. Cells were treated with pterostilbene at increasing concentrations (2.5–60 μM) for 48 h. The obtained results have shown that pterostilbene decreased the incorporation of BrdU into newly synthesized DNA after 48 h in a concentration-dependent pattern (Figure 1). Additionally, this depended on the cell type. A substantial reduction in the proliferative activity of C32 cells was found after incubation with pterostilbene at concentrations ≥ 5 μM. In A2058 melanotic melanoma cells, a significant decrease in the DNA synthesis level was achieved after treatment with higher concentrations of PTB (≥20 μM). No relevant inhibition of cell proliferation was observed in cultures incubated with 2.5 μM PTB. The IC50 values of 21.45 μM and 42.70 μM for C32 and A2058 cells, respectively, reflected their different levels of sensitivity to pterostilbene. Taken together, these experiments revealed that C32 amelanotic cells are more sensitive to pterostilbene than A2058 melanotic cells.

### 2.2. The Influence of Pterostilbene on Melanoma Cell Cycle

The cell cycle distribution of C32 and A2058 melanoma cells was assessed by flow cytometry following PTB treatment at concentrations of 20, 40, and 60 µM for 72 h (Figure 2). It was observed that PTB, in a dose-independent manner, markedly decreased the amount of G1/G0 cells and induced cell cycle arrest at the S phase in amelanotic C32 cells. Furthermore, pterostilbene, at concentrations of 40 and 60 µM, significantly reduced the number of C32 cells in the G2/M phase and increased the sub-G1 fraction of cells (Figure 2a). In A2058 melanotic cells, we observed that treatment with 40 and 60 µM PTB significantly altered the percentage of G1/G0 phase cells compared to the control. Exposure of cells to pterostilbene at all concentrations resulted in a significant increase in the population of cells in the sub-G1 phase, suggesting that it induced cancer cell death (Figure 2b).

### 2.3. The Impact of Pterostilbene on Transcriptional Activity of Genes Encoding the Cell Cycle-Regulating Proteins

The *CCND1* and *CDKN1A* genes encode key cell cycle proteins, i.e., cyclin CD1 and p21^Waf1/Cip1^, respectively. Cyclin D1 determines the cell’s transition from the G1 phase to the S phase of the cell cycle, and the p21 protein acts as an inhibitor of cyclin-dependent kinases, resulting in the course of cell cycle inhibition. Therefore, we determined the expression of *CCND1* and *CDKN1A* mRNAs in melanoma cells treated with 20, 40, and 60 µM. Exposure of amelanotic C32 melanoma cells to pterostilbene at all concentrations for 12 h resulted in down-expression of *CCND1* mRNA compared to untreated cells (Figure 3a). However, the obtained results demonstrated that PTB had no statistically important influence on this gene expression in melanotic A2058 cells (Figure 3b).

Compared to the control, an increase in the *CDKN1A* mRNA expression was detected in both melanoma cell lines treated with PTB for 12 h. However, in the C32 culture, a stronger effect of PTB action than on A2058 was observed (Figure 3). The increase in the transcriptional activity of the *CDKN1A* gene in both amelanotic and melanotic melanoma cells under the influence of pterostilbene may result in cell cycle arrest and a decrease in the proliferation potential of melanoma cells.

### 2.4. The Effect of Pterostilbene on the p21^Waf1/Cip1^ Protein Level

To establish whether the increase in *CDKN1A* mRNA expression corresponded with p21^Waf1/Cip1^ protein level at the next step of the study, the influence of pterostilbene on the concentration of p21 protein in melanoma cells was evaluated (Figure 4). Amelanotic C32 cells characterized higher p21 protein levels than melanotic A2058 cells. The obtained results revealed that PTB had no effect on the p21 concentration in C32 cultures. Exposure of A2058 cells to PTB at all concentrations for 24 h resulted in up-expression of the p21 protein compared to untreated cells. These findings indicated a pterostilbene effect on the *CDKN1A* gene in melanotic A2058 cells, both at the mRNA and protein levels.

### 2.5. The Influence of Pterostilbene on Melanoma Cell Apoptosis

To further examine whether the mechanism of action of pterostilbene in amelanotic and melanotic melanoma cells involves the induction of apoptosis, the influence of pterostilbene on the expression of genes encoding apoptosis-related proteins (BAX, caspases 3 and 9), caspase 3 activity, and DNA fragmentation levels was evaluated. The transcriptional activity of the examined genes was analyzed in control cells and cells treated with different concentrations of pterostilbene for 12 h using quantitative RT-PCR (Figure 5). BAX belongs to the BCL2 protein family and functions as an apoptotic activator that determines the survival or death of cells. The experimental data revealed that pterostilbene was found, at all concentrations, to significantly increase the expression of the *BAX* gene in amelanotic C32 cells in relation to the controls (Figure 5a). However, treatment of melanotic cells with pterostilbene negatively regulated *BAX* mRNA expression (Figure 5b).

In the next step, the transcription level of the genes encoding initiator caspase 9 (*CASP9*) and the executive caspase 3 (*CASP3*) were analyzed. Compared to the control, the expression of the *CASP9* and *CASP3* mRNAs in amelanotic C32 cells showed that pterostilbene, at all concentrations, significantly increased the transcriptional activity of both genes by approximately two-fold. However, the strongest effect was observed in cells exposed to PTB at a concentration of 40 μM (Figure 5a). The results indicated that pterostilbene had no influence on the expression of either *CASP9* or *CASP3* mRNAs in A2058 cells (Figure 5b).

Caspase 3 is an enzyme that plays a crucial role in the execution phase of cell apoptosis. To determine the influence of pterostilbene on the induction of caspase 3 activity, cells were incubated with pterostilbene at concentrations of 20, 40, and 60 μM for 48 and 72 h. The effect of pterostilbene on caspase 3 activity in melanoma cells is presented in Figure 6. Pterostilbene, at all concentrations, statistically increased caspase 3 activity in amelanotic C32 cells after 48 and 72 h. The strongest effect was observed in cells exposed to pterostilbene at a concentration of 40 μM at both time points. On the contrary, pterostilbene did not influence caspase 3 activity in melanotic A2058 cells, which may indicate the inability of pterostilbene to induce apoptosis of these cells. Thus, these data suggest that active caspase 3 could be involved in the induction of apoptosis by pterostilbene in amelanotic C32 cells, but not in melanotic A2058 cells.

To confirm the apoptotic activity of pterostilbene in melanoma cells, a Cell Death Detection ELISA^PLUS^ assay was performed. Cultures C32 and A2058 were incubated with pterostilbene for 24 and 48 h. After 24 h, statistically significant 6.7-fold, 7.6-fold, and 6.6-fold increases in enrichment factors were observed in cells treated with 20, 40, and 60 µM pterostilbene, respectively. Longer incubation (72 h) of C32 with this stilbene caused a similar effect on released nucleosomes (Figure 7). Our results demonstrated that pterostilbene did not influence the formation of nucleosomes in A2058 cells.

These findings may suggest that pterostilbene induces caspase-dependent apoptotic death of amelanotic C32 cells, but does not stimulate this pathway in melanotic A2058 melanoma cells. The different effects of this stilbene on melanoma cells may depend on melanin pigmentation of cells and the difference in the origin of the tested cells.

## 3. Discussion

Melanoma is a serious medical problem. Relatively high mortality and a growing number of newly diagnosed cases result in a growing need for research on melanoma. Treatment of advanced melanoma is supported by chemotherapy, immunotherapy, or radiotherapy, but its anticancer effects are unsatisfactory [19]. Therefore, novel and more effective treatment strategies need to be developed.

Phytochemicals are natural compounds that have gained attention as promising chemopreventive and chemotherapeutic agents due to studies that have demonstrated their ability to prevent the development of skin cancer [20,21]. Different phytochemicals perform various functions, including inducing the death of cancer cells by arresting the cell cycle and inhibiting angiogenesis or metastasis [22]. Among the different classes of phytochemicals, polyphenols are promising in the treatment of melanoma [20]. One of the most studied polyphenols is resveratrol. Several preclinical studies have revealed that resveratrol is active against melanoma; it has been shown to decrease the growth of amelanotic and melanotic melanoma cells [23]. Furthermore, it has been shown to induce the death of melanoma cells through caspase-dependent and caspase-independent pathways [24,25]. Despite its promising anticancer role, resveratrol has unfavorable pharmacodynamics due to its high metabolism leading to a reduced concentration in the human body [26]. Therefore, much focus has shifted towards other polyphenols with biological activity. Pterostilbene is a naturally occurring polyphenol found in blueberries and grapes that shows a higher level of bioavailability than resveratrol [27]. It has gained increasing attention due to its role in the prevention of different diseases, such as cardiovascular diseases, neurological disorders, and metabolic diseases [28,29,30]. Pterostilbene has previously been described to have promising anticancer activity against various types of cancer by influencing cancer cell proliferation and death, among other effects [31]. It has been reported to arrest cell-cycle progression at the G1/G0 phase in breast cancer cells by up-regulation of p21 and cyclin D1 suppression [32]. Furthermore, the antiproliferative and cell death-inducing activity of pterostilbene has been shown against pancreatic [33], colon [34], lung [35], and ovarian [36] cancer cells in vitro and in vivo. Despite reports on the anticancer properties of pterostilbene, data on its activity against melanoma cells are still limited. Therefore, it is important to investigate the effect of pterostilbene on melanoma cells.

Since evaluation of cancer cell proliferation and death is a key component in the discovery and development of antineoplastic drugs in the current study, changes in the growth of two different human melanoma cells, melanotic A2058 and amelanotic C32, were studied after incubation with pterostilbene, as was the ability of this stilbene to induce cell death. In more detail, the results of our study revealed a significant dose-dependent reduction in the proliferation rate of both types of melanoma cells after incubation with pterostilbene. The study by Bennloch et al. [37] also revealed that pterostilbene, at low concentrations (1–5 µM), did not alter the growth nor the viability of melanoma cells (A2058, MeWo, and MelJuso) after 72 h of incubation. The time- and dose-dependent growth-inhibitory effect of pterostilbene was observed at concentrations of 10–100 μM against A375 melanoma cells, as well as lung cancer cells (A549), colon cancer cells (HT29), and breast cancer cells, in studies carried out by Mena et al. [38]. The obtained value of pterostilbene’s half-maximal inhibitory concentrations for A549 cells was 14.7 µM, and was lower than that for breast cancer cells (44 µM) and colon cancer cells (60.3 µM). Due to the different sensitivities of the cells studied to pterostilbene, the authors suggested that the reduction in the number of tumor cells depends on differences in the cell lines assayed.

Cell cycle analysis was performed to examine the mechanism underlying the effect of pterostilbene on melanoma cells. The analysis performed herein indicated that pterostilbene caused changes in the cell cycle profile of both C32 and A2058 cells; however, the influence appeared to be different depending on the type of cells. Pterostilbene decreased the number of cells in the G1/G0 and G2/M phases, and also induced cell cycle arrest at the S phase in amelanotic C32 cells. Treatment of melanotic cells with pterostilbene also reduced the percentage of cells in the G1/G0 phase, but the effect was weaker than in amelanotic cells. In both cell lines, an increase in cell population in the sub-G1 phase was observed after exposure to pterostilbene. Therefore, these results indicate the ability of pterostilbene to induce the arrest of the melanoma cell cycle and the induction of cell death. Although many studies have revealed the influence of pterostilbene on the cancer cell cycle and changes in its distribution in various phases, so far only single studies have been published on its effect on melanoma cells. It has been found that the mechanism of action of pterostilbene varies in different cells. Pterostilbene led to inhibition of the cell cycle in the S phase with a concomitant decrease in the number of cells in the G2/M phase in a SMMC-7721 hepatocellular carcinoma [39]. The results of these studies are consistent with the data obtained in the present study for C32 melanoma cells. The accumulation of cells in the S phase of the cell cycle may result from the activation of DNA repair processes. In lymphoma cells, pterostilbene was found to increase the level of CHEK2, a protein kinase known as an important mediator of the DNA damage checkpoint for phosphorylate proteins involved in DNA repair and cell cycle arrest [40,41]. Chen et al. [42] showed a pterostilbene-induced increase in a marker protein of double-stranded DNA breaks contributing to cells’ genomic instability in H929R myeloma cells. They suggested that pterostilbene could induce DNA damage leading to cell cycle arrest in the S phase. In oral cancer cells SAS and OECM-1, pterostilbene caused both S phase cycle arrest and increased the percentage of the cell population in the G1/G0, which may indicate that arrest of cell division was inhibited in the G1 phase [43]. Pterostilbene also induced cell cycle arrest in the G1 phase in HL-40 leukemia. However, in the case of these cells, it also caused a decrease in the number of cells in the S and G2/M phases [44]. The effect on pterostilbene on melanoma cells was only analyzed by Mena et al. [38]. They revealed that a high concentration of pterostilbene (75 µM) induced the portion of A375 cells in the G1 phase. However, pterostilbene, at lower concentrations, reduced the number of A375 melanoma cells in the G1 phase with an increase in the population of cells in the S phase. These results are in line with the findings of the current study. The presented studies suggest that the effect of pterostilbene on the cell cycle varies depending on the concentration used, the origin of the cells, and their invasiveness. Therefore, further experiments should be conducted to determine the exact mechanism of pterostilbene activity in melanoma cells.

To clarify the mechanism of pterostilbene activity against colon cancer cells, we also evaluated changes in the expression of the cell cycle regulatory genes *CCND1* and *CDKN1A*. Molecular analysis of human melanoma reveals that cell cycle regulators are frequently mutated. The frequency of mutations in the *CCND1* gene encoding cyclin D1 depends on the melanoma subtype and ranges from 19 to 80%. Moreover, CCND1/cyclin D1 up-regulation favors the growth and development of the primary tumor; therefore, it is considered as the proto-oncogene [45]. The *CDKN1A* gene encodes the protein p21, which acts as an inhibitor of cyclin-dependent kinases (CDKs). It not only promotes cell cycle arrest, but also interacts with the proliferating cell nuclear antigen (PCNA), leading to the arrest of DNA synthesis and the initiation of repair processes [46]. In melanoma cells, disturbances in *CDKN1A* expression are frequent, and usually lead to their uncontrolled proliferation [47]. Cyclin D and the p21 protein participate in the regulation of the cell cycle at the G1/S restriction point. Our data indicated that treatment of C32 amelanotic melanoma cells with pterostilbene significantly decreased gene expression in C32, suggesting disruption of the uncontrolled progression of the cell cycle of these cells. Surprisingly, pterostilbene did not significantly affect *CCND1* transcriptional activity in A2058 melanotic melanoma cells; therefore, more studies are required to determine the antiproliferative mechanism of pterostilbene against these cells. Interestingly, there is a lack of RB protein that is regulated by cyclin D in A2058 cells. The loss of the functional RB protein was associated with the failure of cells to arrest the cell cycle at the G1/S checkpoint [48]. Until now, the expression of cyclin D in pterostilbene-treated melanoma cells has not been evaluated. However, published studies have shown the ability of pterostilbene to down-regulate cyclin D in several types of cancer cells, such as breast cancer [32] and colon cancer [36] cells. Furthermore, pterostilbene increased the expression of the *CDKN1A* gene that encodes the p21 protein in amelanotic and melanotic cells. However, changes in *CDKN1A* mRNA were accompanied by up-regulation of p21 expression only in melanotic cells. Based on data from the literature, increased expression of p21 blocks the transition of cells from the G1 to the S phase. Similar results were obtained in the present study. Incubation of A2058 cells with pterostilbene resulted in an increase in the p21 protein; however, it did not influence the number of cells in phase S. In C32 cells, pterostilbene did not affect the concentration of the p21 protein and did not increase the number of cells in S phase [49]. In this context, it can be suggested that one of the possible mechanisms of the biological activity of pterostilbene is related to the inhibition of proliferation. The arrest cell cycle in the G1 phase is an opportunity for cells to go through the repair mechanism or follow the apoptosis pathway.

As observed, pterostilbene revealed a strong growth-inhibitory effect. Incubation of melanoma cells with pterostilbene caused a reduction in DNA synthesis and cell cycle arrest, as well as a significant decrease in the number of cells, which can only be explained when cell death is induced. Pterostilbene has been shown to activate cancer cell death through different mechanisms such as apoptosis, autophagy, and necrosis; however, apoptosis is assumed to be the main type of pterostilbene-induced cell death [50,51,52]. Apoptosis plays a crucial role in the elimination of damaged and abnormal cells, and dysregulation in the apoptotic cell death machinery is a hallmark of a variety of cancer cells [53]. Melanoma cells are characterized by a low sensitivity to apoptosis associated with high expression of antiapoptotic proteins of the BCL-2 family [7], loss of activity of Apaf-1 factor [54] and up-regulation of the PI3K/AKT/mTOR pathway [55]. Therefore, apoptosis-targeted drugs are promising in melanoma therapy. To determine the mechanism of pterostilbene-induced cell death in the current study, the possible impact of pterostilbene on *BAX*, *CASP9*, and *CASP3* gene expression was evaluated. We found that pterostilbene significantly increased the expression of *BAX*, *CASP9*, and *CASP3* in amelanotic C32 cells, but not in melanotic A2058 cells. Apoptosis is tightly regulated by effector molecules such as executioner caspase 3, which is responsible for the breakdown of the nucleus and other cellular compartments during apoptosis [56]. The current study revealed that pterostilbene activated caspase 3 in C32 cells, but not in A2058 cells. To confirm the pro-apoptotic influence of pterostilbene, we investigated nucleosome formation, a marker of apoptotic death. The results showed that pterostilbene significantly increased nucleosome occurrence in amelanotic C32 cells, but had no effect in melanotic A2058 cells, indicating different mechanisms of cell death-inducing activity of pterostilbene in C32 and A2058 melanoma cells. Based on the current results, we confirmed that pterostilbene activated caspase-dependent apoptosis in human melanoma amelanotic C32 cells, as well as caspase-independent cell death in melanotic A2058. This implies that pterostilbene could induce cell death through various molecular pathways. Despite limited reports on the pro-apoptotic effects of pterostilbene on melanoma cells, numerous studies have shown multidirectional mechanisms of pterostilbene-induced cell death in breast, gastric, prostate, colon, and lung cancer cells [57,58,59,60,61]. Pterostilbene has been shown to cause depolarization of mitochondrial membranes, release of cytochrome C [59,62], up-regulation of the expression of pro-apoptotic proteins, increases in caspase activity, and inhibition of the expression of anti-apoptotic proteins [63]. Research evidence suggests that pterostilbene, dependent on the cell type, can induce different mechanisms of cell death. Studies by Mena et al. [38] showed that pterostilbene significantly induced caspase 3 activity in melanoma A375 cells and lung cancer A549 cells; however, it did not have this effect in colon HT-29 cancer cells or breast MCF-7 cancer cells. Studies carried out by Ferrer et al. [64] found that the pro-apoptotic properties of pterostilbene in B16M-F10 melanoma cells were indicated by an increase in expression of the *BAX* gene and inhibition of expression of the *BCL-2* gene. Pterostilbene has also been reported to induce the formation of apoptotic bodies and increase the activity of caspases 3 and 7 in SK-MEL-2 melanoma cells [52]. Furthermore, pterostilbene has also been reported to induce necrosis, cancer cell autophagy [50], and permeabilization of the lysosomal membrane [38].

Overviewing all of our data together, pterostilbene demonstrates a potent anticancer influence against melanoma cells in vitro thought cyclin D1, p21, and caspase 3, causing cell cycle arrest as well as both caspase-dependent and caspase-independent cell death (Figure 8).

Furthermore, our results showed significant differences between the studied C32 amelanotic and A2058 melanotic cells. As observed, a more pronounced effect of pterostilbene was detected against amelanotic cells than against A2058 cells. Moreover, in amelanotic cells, it induced caspase-dependent apoptotic cell death.

Despite significant progress in our understanding of melanoma, its heterogeneity is one of the major obstacles to clinical efficiency of anticancer drugs. Melanoma cells are known for their high plasticity and ability to switch back and forth between different melanoma cell states [65]. Heterogeneity can be observed at both the genetic and biological levels, even within melanoma cell lines. Biological heterogeneity includes switching between mostly melanotic proliferative and invasive mesenchymal-like states of cells. Different phenotypes may also have different sensitivities to a drug. Moreover, some melanoma cells that are responsive to drugs can evolve into a drug-tolerant state [65,66]. The consequences of the highly heterogeneous nature of melanoma are reflected in the clinical presentation of the therapeutic response, and represent a challenge for identifying effective treatment strategies. The sensitivity of melanoma cells to targeted inhibitors and immunotherapies has been shown to be associated with different levels of cancer cell differentiation (differentiated phenotypes exhibit higher sensitivity to BRAF inhibitors) [67,68]. Furthermore, cells with innate sensitivities to BRAF inhibitors may become drug-resistant as a result of BRAF inhibition [65]. Recently published studies have shown that even isogenic melanoma cells can undertake different, independent trajectories between drug-responsive and drug-tolerant states, each of them characterized by unique metabolic and signaling networks [66]. New methods, such as single-cell functional proteomics, allow the identification of signaling pathways activated after BRAF inhibition and before the appearance of drug-resistant phenotypes. That cell-to-cell differences seem critical to therapeutic response to single-agent therapies, and simultaneously, targeting these distinct pathways is essential to fully prevent drug tolerance. Combination therapy using drugs targeting those pathways with BRAF inhibition may be able to halt the adaptive transition [67,68]. Therefore, it is important to incorporate intratumor heterogeneity and the expected evolutionary trajectories to drug tolerance into the design of a new drug combination for melanoma treatment. Studies also showed the relationship between cellular de-differentiation and metabolic reprogramming. Therefore, it seems interesting to evaluate whether the metabolic differences between cancer cells can be used to resensitize them to drugs. Differentiated melanoma cell lines have been shown to be more sensitive to fatty acid synthesis inhibitors, while differentiated ones are more sensitive to lipid monounsaturation [69]. Understanding the role of heterogeneity in the aggressiveness of melanoma and resistance to treatment could allow us to determine more effective types of therapy; develop new therapeutics, including targeted polypharmacology; and achieve better outcomes for patients [70].

The presented results suggest that pterostilbene is more effective towards the amelanotic C32 cell line. Recognition of the underlying pathways affected by pterostilbene in melanoma cells, as well as the relationship between the mechanism of action of pterostilbene and its ability to synthesize melanin and the mutation profile, seems especially important due to the high heterogeneity of melanoma. It is also essential due to the high aggressiveness of amelanotic melanoma and diagnoses at a more advanced clinical stage than pigmented melanomas. The question still remains as to how to explain the different sensitivities of C32 and A2058 cells to pterostilbene. The melanin content of the C32 and A2058 cells varies. Melanin plays a crucial role in preventing damage caused by free radicals, so melanoma cells with a low melanin content are significantly less resistant to reactive oxygen species damage than cells with higher melanin content [71]. Further investigation of the activity of pterostilbene against amelanotic and melanotic melanoma cells is clearly warranted.

## 4. Materials and Methods

### 4.1. Cell Lines and Cell Culture

The A2058 melanotic and C32 amelanotic human melanoma lines were obtained from the American Type Culture Collection (ATCC, Rockvile, MD, USA). Both melanoma cell lines are highly tumorigenic in nude mice with the BRAFV600E mutation, but differ in their ability to synthesize melanin. Cells were routinely cultured in MEM medium (Sigma Aldrich, St. Louis, MO, USA) supplemented with 10% fetal bovine serum (BioWest, Nualillé, France), 100 U/mL penicillin, and 100 μg/mL streptomycin (Sigma Aldrich) in a humidified atmosphere containing 5% CO_2_ and 95% air at 37 °C. Cells were treated with pterostilbene solutions, and the experiment proceeded as described below.

### 4.2. Preparation of Pterostilbene Solution

The stock solution of pterostilbene (Sigma Aldrich) was prepared by dissolving it in dimethylsulfoxide (DMSO, Sigma Aldrich). Subsequently, pterostilbene solution was diluted in a sterile culture medium to the desired concentration directly prior to use. The final concentration of DMSO was 0.1%.

### 4.3. Cell Proliferation Analysis

To evaluate the effect of pterostilbene on melanoma cell proliferation, a 5-bromo-2′-deoxyuridine (BrdU) colorimetric enzyme-linked immunosorbent assay kit (ELISA) (Roche, Mannheim, Germany) was used. Cells were seeded in 96-well plates at a density of 8 × 10^3^ in 200 µL medium, followed by overnight incubation. Subsequently, the medium was replaced with a fresh one containing pterostilbene (2.5; 5; 10; 15; 20; 40; 60 μM) and treated for 48 h. The BrdU solution was added to the media for the last 4 h of incubation. After removal of the labeling media, cells were fixed and DNA was denatured with FixDenat solution for 20 min. Immune complexes were formed using a peroxidase-conjugated antibody. The incorporation of BrdU into DNA was determined by absorbance measuring at λ = 450 nm (with reference λ = 690 nm) using a microplate spectrophotometer Labtech LT-5000c. The growth of treated cells was expressed as a percentage of untreated control cells. The pterostilbene concentration that reduced proliferation by 50% (IC50) compared to control was determined by fitting a four-parameter logistic model (Hill equation) to the experimental data using computer curve fitting software (GraphPad Prism version 9, San Diego, CA, USA). 

### 4.4. Cell Cycle Analysis

To evaluate the distribution of the cell cycle, A2058 cells (1.5 × 10^5^ cells) and C32 cells (5 × 10^5^ cells) were seeded in 21.5 cm^2^ dishes (Nunc International, Rochester, NY, USA) and grown for 24 h. Subsequently, cell cultures were treated with PTB at concentrations of 0, 20, 40, and 60 µM. After 72 h of incubation, cells were harvested by trypsinization, washed in PBS buffer, and fixed in cold 70% ethanol at −20 °C overnight. Cell pellets were incubated with RNaseA (final concentration, 200 μg/mL) in PBS buffer for 1 h at 37 °C in the dark and then stained with a solution of propidium iodide (final concentration, 10 μg/mL) (Sigma Aldrich). The DNA content and cell cycle distribution of cells were analyzed by a BD FACS Aria II flow cytometer and BD FACSDiva software (BD Biosciences, San Jose, CA, USA).

### 4.5. DNA Fragmentation Assay

To demonstrate the pro-apoptotic potential of pterostilbene “Cell Death Detection, ELISA^PLUS^” (Roche) was used. This method reveals the occurrence of nuclear DNA fragmentation by the use of antibodies against DNA and histones. Melanoma cells were seeded in 96-well plates at a density of 7.5 × 10^3^ cells/200 μL and cultured for 24 h. Then, cells were treated with pterostilbene at concentrations of 20, 40, and 60 μM for 24 h or 48 h. According to the manufacturer’s instructions, cells were lysed and centrifuged to produce a nucleosome-containing supernatant. The enrichment of mono- and oligonucleosomes released into the cytoplasm of cell lysates was detected by biotinylated anti-histone- and peroxidase-coupled anti-DNA-Ab. DNA fragmentation was expressed as the enrichment of histone-associated mono- and oligonucleosomes released into the cytoplasm. Spectrophotometric results were measured at a wavelength of 405 nm. The enrichment factor used as the parameter of apoptosis was calculated using the formula of absorbance of sample cells/absorbance of control cells to estimate the fold increase of DNA fragmentation in treated samples with reference to the control.

### 4.6. Total RNA Extraction and Quantitative Real-Time RT-PCR (RT-qPCR)

To evaluate the expression of the *CCND1*, *CDKN1A*, *CASP3*, *CASP9*, and *BAX* genes, cells were seeded at a density of 8 × 10^5^ onto 21.5 cm^2^ culture plates (Nunc International, Rochester, NY, USA) and grown for 48 h. Then, PTB, at concentrations of 20, 40, and 60 μM, was added to the cell cultures for 12 h. Total RNA was extracted from cells with the use of TRI REAGENT (Zymo Research, Irvine, CA, USA) according to the manufacturer’s instructions. RNA concentration and purity were detected using the Shimadzu UV-1800 spectrophotometer (Shimadzu, Kyoto, Japan). Samples showing a ratio of Abs 260/280 nm between 1.8 and 2.0 were only used for experiments. Detection of the expression of the examined genes was carried out using a RT-qPCR technique with SYBR Green chemistry (SensiFastTM SYBR Green No-ROX One-Step) (Bioline, Meridian Bioscience, Cincinnati, OH, USA) and CFX Connect Real-Time PCR Detection System (Bio-Rad, Hercules, CA, USA). Aliquots (0.1 µg) of total cellular RNA were applied to one-step RT-qPCR at 20 μL reaction volume. Oligonucleotide primers specific for *CASP3* and *CASP9* mRNAs were synthesized in Oligo.pl at the Institute of Biochemistry and Biophysics of the Polish Academy of Sciences (Warsaw, Poland). The primers for *CCND1, CDKN1A,* and *BAX* were commercially available (Sigma-Aldrich). The characteristics of the primers are presented in Table 1. The thermal profile for RT-qPCR was as follows: 45 °C for 10 min for reverse transcription and 95 °C for 2 min, followed by 45 cycles at 95 °C for 5 s, 60 °C for 10 s, and 72 °C for 5 s for amplification. Each gene analysis was performed in triplicate. The mRNA copy numbers of the examined genes were determined on the basis of the commercially available standard of β-actin (TaqMan DNA Template Reagent Kit, Invitrogen, Waltham, MA, USA) and recalculated per µg of total RNA. The expression levels of all genes in cultured cells were expressed as a fold change relative to the corresponding controls. The fold change > 1 and <1 were set as an increased and a decreased expression of the target gene, respectively.

### 4.7. Detection of the p21^Waf1/Cip1^ Protein Level

To determine the concentration of the p21^Waf1/Cip1^ protein, melanoma cells were seeded at a density of 4 × 10^6^ onto 56.7 cm^2^ culture dishes and cultured for 24 h. One day after plating, the cultures were exposed to pterostilbene at concentrations of 20, 40, and 60 μM for 24 h. Afterwards, cells were washed with ice-cold PBS, scrapped from the dishes, centrifuged, and lysed on ice in a cell extraction buffer. Expression of the p21 protein was determined with commercially available ELISA kits (Invitrogen) following the manufacturer’s instructions. The absorbance was measured using the Labtech LT-5000 multiplate reader (Labtech International) at λ = 450 nm. The concentration of p21 was evaluated on the basis of the standard curve generated under identical conditions. The results obtained were normalized to the total protein content in the cells, as measured by the Bradford assay (Sigma-Aldrich).

### 4.8. Caspase 3 Activity

Detection of caspase 3 activity was performed with the commercially available “Colorimetric Caspase 3 Assay Kit” (Sigma Aldrich) based on the hydrolysis of the substrate, Ac-DEVD-pNA (acetyl-Asp-Glu-Val-Asp-pNA), by active caspase 3. Cells were seeded (2.5 × 10^6^) onto 56.7 cm^2^ dishes and cultured for 24 h. Then, cells were treated with pterostilbene at concentrations of 20, 40, and 60 μM for 48 h or 72 h. Subsequently, cells were scraped from the dishes, lysed, and centrifuged. Caspase-3 activity in cell lysates was determined according to the manufacturer’s instructions and normalized to the total content of cellular proteins, which was determined by the Bradford method.

### 4.9. Statistical Analysis

Statistical analysis was performed with the use of Statistica PL ver. 12.0 Software (StatSoft Polska, Cracow, Poland). One-way analysis of variance (ANOVA) with Tukey’s post hoc was used to evaluate significance in the examined group. All data expressed as means ± SD were representative of at least three independent experiments. Values of *p* < 0.05 were considered statistically significant.

## 5. Conclusions

Based on the present study, it may be concluded that pterostilbene exhibits antiproliferative activity in human melanoma cells and induces both caspase-dependent and caspase-independent melanoma cell death. The molecular mechanism of pterostilbene includes the regulation of cyklinD1 and p21 expression, activation of caspase 3, and induction of DNA fragmentation. Furthermore, the results showed significant differences between the studied C32 amelanotic and A2058 melanotic melanoma cells. As observed, a more pronounced effect of pterostilbene was detected against amelanotic. Further research is warranted to fully elucidate the effects of pterostilbene on amelanotic and melanotic melanoma cells.

## Figures and Tables

**Figure 1 ijms-24-01115-f001:**
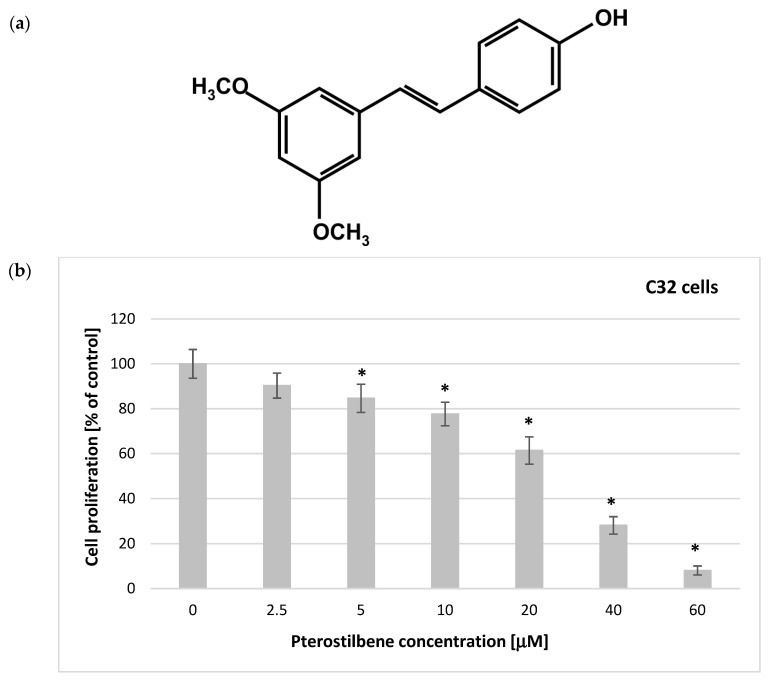
Chemical structures of pterostilbene. (**a**) Influence of pterostilbene on the proliferation of (**b**) C32 and (**c**) A2058 cells after 48 h of treatment. The results are expressed as a percentage of the untreated control (means ± SD; * *p* < 0.05 vs. control).

**Figure 2 ijms-24-01115-f002:**
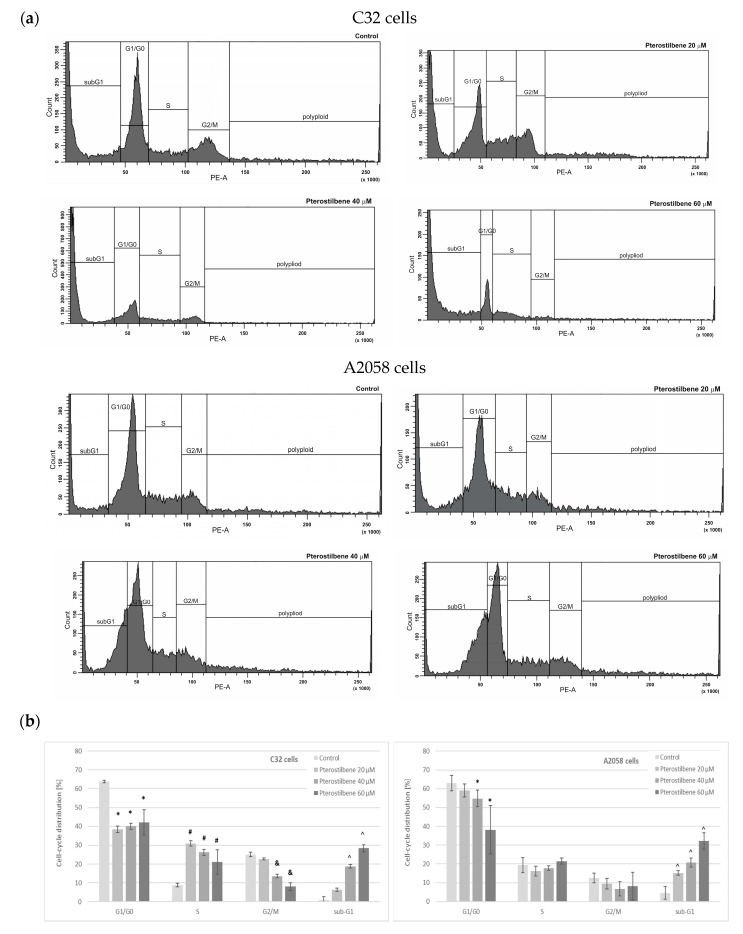
The effect of pterostilbene on the C32 and A2058 cell cycle after 72 h. The cells were labeled with propidium iodide for DNA contents and analyzed by flow cytometry. (**a**) Representative histograms of cell cycle analysis. (**b**) Cell cycle distribution. The data indicate the percentage of cells in each phase of the cell cycle. (means ± SD; * (G1/G0), # (S phase), and & (G2/M), ^ (subG1), *p* < 0.05 vs. control).

**Figure 3 ijms-24-01115-f003:**
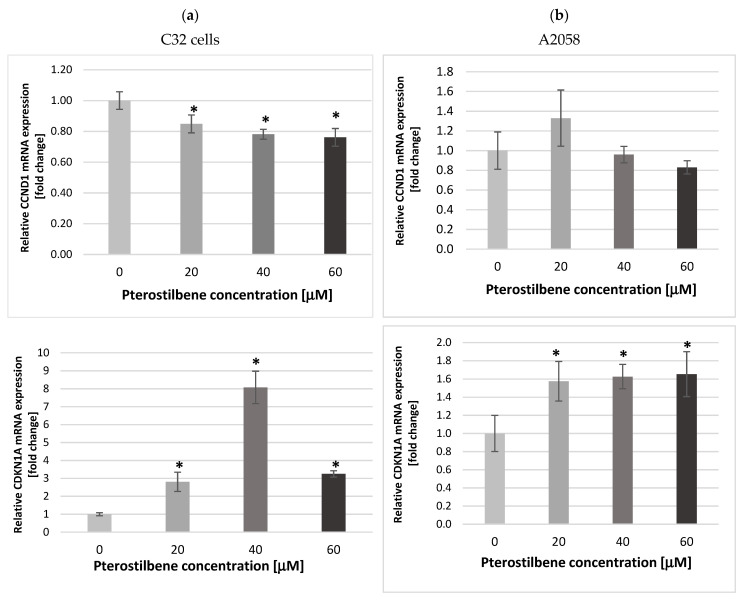
Expression of *CCND1* and *CDKN1A* mRNAs in (**a**) C32 and (**b**) A2058 melanoma cells treated with 20, 40, and 60 µM pterostilbene for 12 h. The results are presented as mean ± SD; * *p* < 0.05 vs. the control.

**Figure 4 ijms-24-01115-f004:**
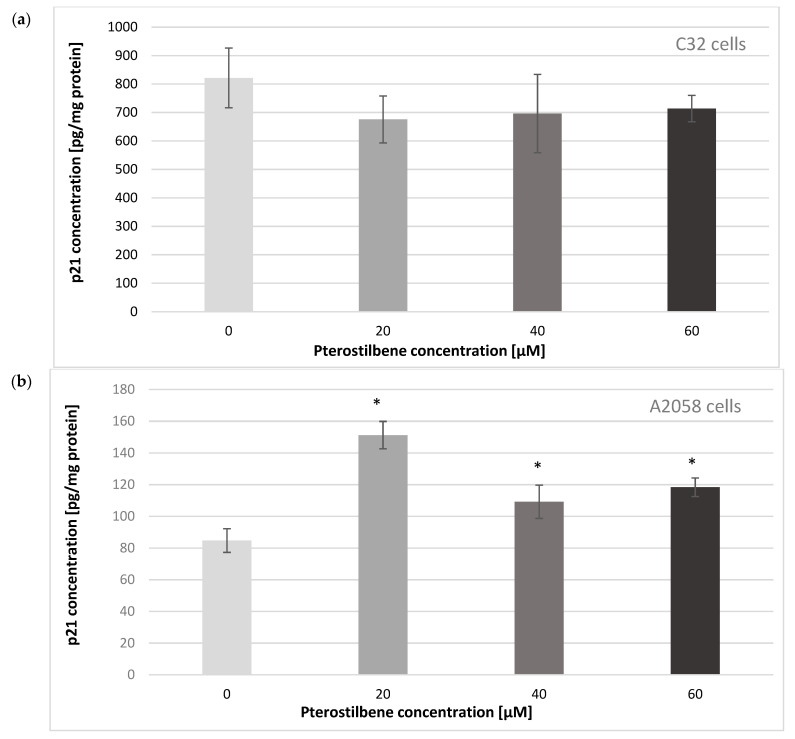
Effect of pterostilbene at concentrations of 20, 40, and 60 µM on the p21 protein in (**a**) C32 and (**b**) A2058 melanoma cells at 24 h. The results are presented as mean ± SD; * *p* < 0.05 vs. control.

**Figure 5 ijms-24-01115-f005:**
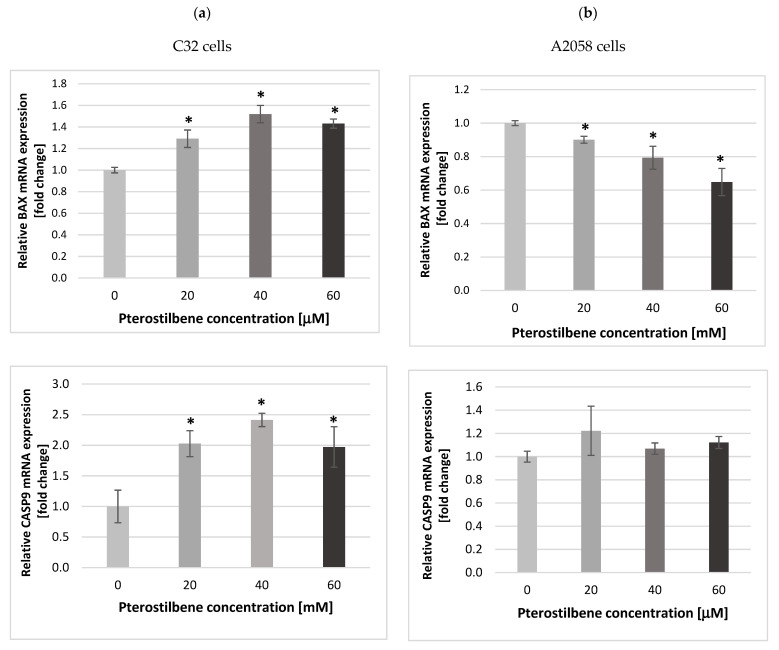
Expression of the *BAX, CASP9* and *CASP3* genes in (**a**) C32 and (**b**) A2058 cells after 12 h of incubation with pterostilbene. The results are presented as mean ± SD; * *p* < 0.05 vs. control.

**Figure 6 ijms-24-01115-f006:**
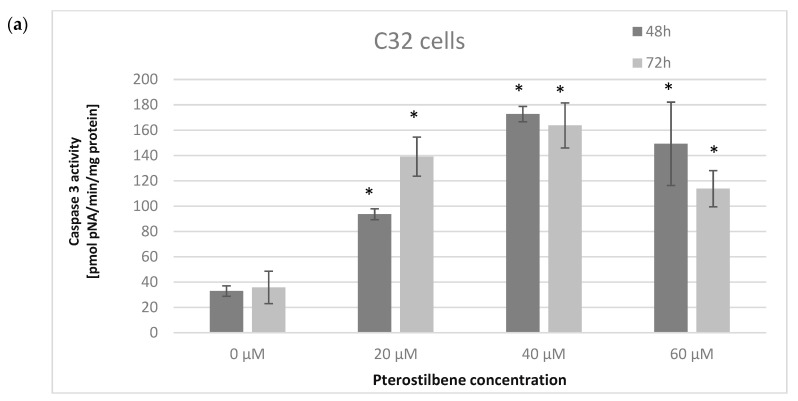
Effect of pterostilbene on caspase 3 activity in (**a**) C32 and (**b**) A2058 cells after 48 and 72 h of incubation. The results are presented as mean ± SD; * *p* < 0.05 vs. control.

**Figure 7 ijms-24-01115-f007:**
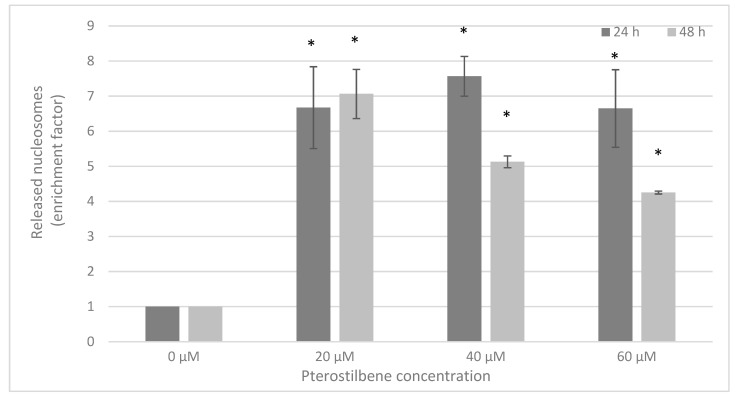
Effect of pterostilbene on the induction of apoptosis in C32 cells evaluated by the mean DNA fragmentation after 24 h and 48 h of incubation. Data are expressed as enrichment factors (means ± SD; * *p* < 0.05 vs. control).

**Figure 8 ijms-24-01115-f008:**
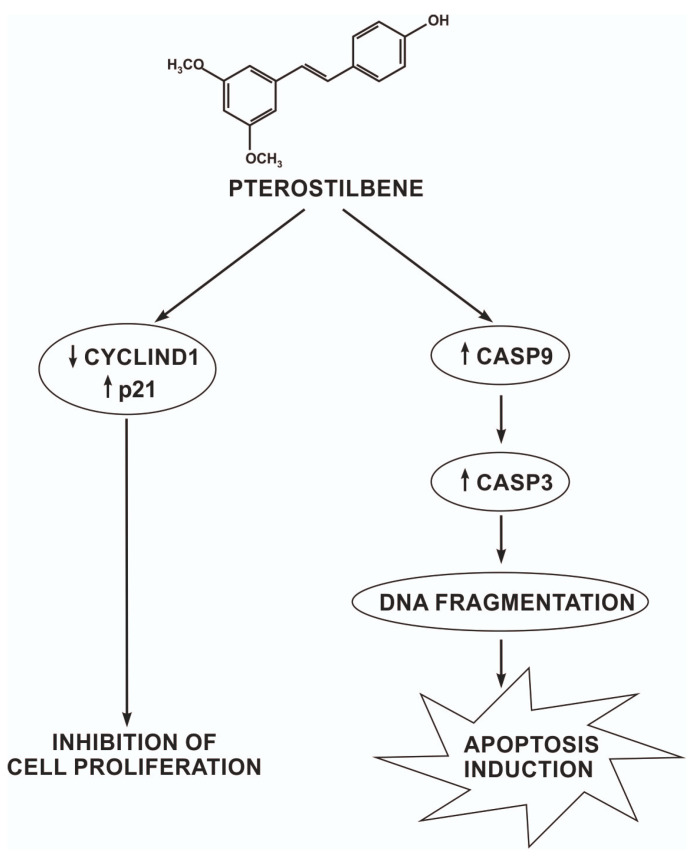
The potential mechanism of pterostilbene action on melanoma cells.

**Table 1 ijms-24-01115-t001:** Characteristics of the primers used in the experiment.

Gene	Forward Primer (5′-3′)	Reverse Primer (5′-3′)
*CCND1*	GCCTCTAAGATGAAGGAGAC	CCATTTGCAGCAGCTC
*CDKN1A*	AGGGATTTCTTCTGTTCAGG	GACAAAGTCGAAGTTCCATC
*CASP3*	GGCCTGCCGTGGTACAGAACTGG	AGCGACTGGATGAACCAGGAGCCA
*CASP9*	GACCGGAAACACCCAGACCAGTGGA	GCAGTGGCCACAGGGCTCCAT
*BAX*	TCTGAGCAGATCATGAAGAC	TCCATGTTACTGTCCAGTTC

## Data Availability

Data are contained within this article.

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
