# Peer review of "Pterostilbene-Mediated Inhibition of Cell Proliferation and Cell Death Induction in Amelanotic and Melanotic Melanoma"

_ijms, 2023, doi:10.3390/ijms24021115_

Round 1
Reviewer 1 Report
Reviewer 1
Research article on Pterostilbene-mediated inhibition of cell proliferation and cell death induction in amelanotic and melanotic melanoma by Joanna Wawszczyk et al. can be considered for publication after major revision.
1. Introduction part should be mentioned with recent references.
2. There are many mistakes in abstract. This part of the manuscript can be improved.
3. Authors shown that the results indicated that pterostilbene ad no influence on the expression of both CASP3 and CASP3 mRNAs in A2058 cells. Is there any effect of pterostilbene on p53 if authors have studied include the details.
4. Following part of the manuscript has to be referenced.
The broad spectrum of biological activity of pterostilbene suggests that this compound may have a preventive and therapeutic effect on malignant melanoma. To date, there are only a few reports on the biological activity of pterostilbene in melanoma cells and the mechanism by which pterostilbene can exert a potential anticancer effect on skin cancers. Most anticancer drugs exert their biological activity by inhibiting the cell cycle and induction of programmed cell death, such as apoptosis, autophagy, and necroptosis. Therefore, the objective of this study was to evaluate the influence of pterostilbene in a wide range of concentrations on the proliferation and apoptosis of human amelanotic and melanotic melanoma cells in vitro.
5. Whether authors have carried virtual screening of the bioactive compounds if not what is the base for the study.
6. There are many mistakes in the manuscript; for example:
However, to confirm the effectiveness of pterostilbene in melanoma treatment more research is steel needed. Correct this sentence.
7. Whether authors have carried toxicity study of compound or not please mention the details.
8. Conclusion part has to be modified for better understanding.
9. Obviously, quality of the manuscript will further increase if authors include structure of pterostilbene and molecular mechanism for the inhibitory potential of the molecule.
10. Authors have studied any enzyme specific inhibition by pterostilbene? For details you may refer following article.
Modak R, Basha J, Bharathy N, Maity K, Mizar P, Bhat AV, Vasudevan M, Rao VK, Kok WK, Natesh N, Taneja R, Kundu TK. Probing p300/CBP associated factor (PCAF)-dependent pathways with a small molecule inhibitor. ACS Chem Biol. 2013;8(6):1311-23. doi: 10.1021/cb4000597.
11. Please mention standard used for the study.
12. Authors are informed to upload original images in form of supporting information for further clarifications.
13. To study effect on p21 by pterostilbene that cause apoptosis can also be investigated using an Annexin V-PE/7-AAD Apoptosis Detection Kit and MTT assay. These assays can further support the confirmation of submitted research work.
Reference:
Guo Q, Wang D, Liu Z, Li C. Effects of p21 Gene Down-Regulation through RNAi on Antler Stem Cells In Vitro. PLoS One. 2015 Aug 26;10(8):e0134268. doi: 10.1371/journal.pone.0134268. PMID: 26308075; PMCID: PMC4550451.
14. The effect of pterostilbene on the expressions of caspase-3, caspase-9, Bcl-2 and Bax can be studied using western blot for further confirmation.
Reference:
Zhao Y, Jing Z, Lv J, Zhang Z, Lin J, Cao X, Zhao Z, Liu P, Mao W. Berberine activates caspase-9/cytochrome c-mediated apoptosis to suppress triple-negative breast cancer cells in vitro and in vivo. Biomed Pharmacother. 2017 Nov;95:18-24. doi: 10.1016/j.biopha.2017.08.045.
Please upload original images in form of supporting information for further clarifications
Author Response
Response to the Reviewer’s Comments
The authors thanks the Reviewer #1 for his (her) detailed revision and valuable comments. We completed and corrected the manuscript according to the Reviewer’s suggestions.
- Introduction part should be mentioned with recent references.
Response:
As suggested by the Reviewer, the references in the Introduction part has been was replaced with most recent.
- There are many mistakes in abstract. This part of the manuscript can be improved.
Response:
We corrected this part according to the Reviewer`s recommendation.
- Authors shown that the results indicated that pterostilbene ad no influence on the expression of both CASP3 and CASP3 mRNAs in A2058 cells. Is there any effect of pterostilbene on p53 if authors have studied include the details.
Response:
In our manuscript, we focus on the influence of pterostilbene on the transcriptional activity on CCND1, CDKN1A, BAX, CASP3, CASP9 genes, p21 protein expression, and caspase 3 activity. Unfortunately, we did not perform an analysis of the p53 protein. We agree with the Reviewer that it would be interesting to explore this aspect, and certainly it would increase the value of our research. To the best of our knowledge, there have been no published studies on the expression of the p53 protein in pterostilbene-treated melanoma cells. However, pterostilbene has the ability to upregulate the p53/p21 pathway in squamous cell carcinoma model [1]. The P53 protein is a central tumor suppressor that activates the transcriptional activity of the CDKN1A gene. Following activation of p53 p21 expression is upregulated. The high levels of p21 then result in downregulation of a large number of cell cycle genes. The reduced expression of the many regulators leads to cell cycle arrest [2]. In our studies, we have shown that the expression of gene encoding the p21 protein, as well as p21 protein level, was elevated in A2058 cells exposed to pterostilbene. Simultaneously, no statistically significant changes in caspase 3 activity (a key downstream executive molecule for apoptosis) and DNA fragmentation were observed. Based on these results, we concluded that pterostilbene caused caspase-independent death in A2058 cells.
[1] Surien, O., Ghazali, A. R., & Masre, S. F. (2021). Chemopreventive effects of pterostilbene through p53 and cell cycle in mouse lung of squamous cell carcinoma model. Scientific reports, 11(1), 14862. https://doi.org/10.1038/s41598-021-94508-7
[2] Engeland K. (2022). Cell cycle regulation: p53-p21-RB signaling. Cell death and differentiation, 29(5), 946–960. https://doi.org/10.1038/s41418-022-00988-z
- Following part of the manuscript has to be referenced.
The broad spectrum of biological activity of pterostilbene suggests that this compound may have a preventive and therapeutic effect on malignant melanoma. To date, there are only a few reports on the biological activity of pterostilbene in melanoma cells and the mechanism by which pterostilbene can exert a potential anticancer effect on skin cancers. Most anticancer drugs exert their biological activity by inhibiting the cell cycle and induction of programmed cell death, such as apoptosis, autophagy, and necroptosis. Therefore, the objective of this study was to evaluate the influence of pterostilbene in a wide range of concentrations on the proliferation and apoptosis of human amelanotic and melanotic melanoma cells in vitro.
Response:
As suggested by the Reviewer we introduced relevant references. Obviously, the list of references has been updated.
- Whether authors have carried virtual screening of the bioactive compounds if not what is the base for the study.
Response:
Our studies focus on the exploring the effects of dietary phytochemicals on various human cancer cells. We did not perform a virtual screening of bioactive compounds. The published studies have drawn our attention to resveratrol due to its wide range of health beneficial properties. Moreover, this stilbene has become popular in connection with the “French Paradox”. It has been shown to exert antioxidant, anti-inflammatory, chemopreventive and anti-aging effects and is potentially capable of inhibiting carcinogenesis at the stages of initiation, promotion and progression. However, despite the many valuable properties, resveratrol has been found to have low bioavailability and a short half-life in vivo. Therefore, we decided to focus our attention on a compound with a structure similar to resveratrol, but with better pharmacokinetic parameters. Pterostilbene is a natural methoxylated resveratrol derivative. It has a higher biostability as a result of slower metabolism and a lower excretion rate compared to other stilbenes. Due to its structural similarity to resveratrol, pterostilbene has resveratrol-like health benefits. So we decided to explore the anticancer properties of pterostilbene against cancer cells. To date, there are many studies regarding pterostilbene activity against various cancers. However, only a few reports are concerned with the biological activity of pterostilbene in melanoma cells and the mechanism by which pterostilbene can exert potential anticancer effects in skin cancers. Therefore, we decided to focus on the effects of pterostilbene on melanotic and amelanotic melanoma cells.
- There are many mistakes in the manuscript; for example:
However, to confirm the effectiveness of pterostilbene in melanoma treatment more research is steel needed. Correct this sentence.
Response:
We checked the overall manuscript for mistakes and corrected errors.
- Whether authors have carried toxicity study of compound or not please mention the details.
Response:
The design of the experiments focused firstly on investigating the cytotoxic effect of pterostilbene on melanoma cells. To ascertain the toxicity of pterostilbene on amelanotic and melanotic melanoma cells we performed the Sulforhodamine B (SRB) assay which is commonly used test to measure drug-induced cytotoxicity (data not shown in the manuscript). The design of the experiments focused first on investigating the response of pterostilbene-treated melanoma cells. Cells were treated with pterostilbene at increasing concentrations (2.5-60 μM) (Fig. 1). The influence of the lowest pterostilbene concentration (2.5 μM) on amelanotic and melanotic was not significant. The statistically significant cytotoxicity of both cell lines was evoked by higher concentrations of pterostilbene (≥ 5 μM). Pterostilbene at a concentration of 5 μM caused significant viability suppression of C32 and A2058 cells by 13% and 6%, respectively. At pterostilbene concentrations ≥ 10 μM, the much stronger effect was observed in amelanotic C32 cells than in melanotic A2058 cells. The most pronounced reduction of C32 and A2058 cell viability in relation to control (98.5% and 81.6% respectively) was observed in cultures exposed to pterostilbene at the highest concentration.
|
A |
|
B |
|
Fig. 1 Cytotoxic effect of pterostilbene on (A) C32 and (B) A2058 cells after 60 hours of treatment (the means ± SD; * p < 0.05 vs. control).
- Conclusion part has to be modified for better understanding.
Response:
As suggested by the Reviewer we have modified the Conclusion part.
- Obviously, quality of the manuscript will further increase if authors include structure of pterostilbene and molecular mechanism for the inhibitory potential of the molecule.
Response:
According to the recommendation of the Reviewer we included in the manuscript the structure of pterostilbene and the schema which demonstrated molecular mechanism of pterostilbene activity on melanoma cells.
- Authors have studied any enzyme specific inhibition by pterostilbene? For details you may refer following article.
Modak R, Basha J, Bharathy N, Maity K, Mizar P, Bhat AV, Vasudevan M, Rao VK, Kok WK, Natesh N, Taneja R, Kundu TK. Probing p300/CBP associated factor (PCAF)-dependent pathways with a small molecule inhibitor. ACS Chem Biol. 2013;8(6):1311-23. doi: 10.1021/cb4000597.
Response:
The prepared manuscript is the first that concerns the influence of pterostilbene on cellular processes in melanoma cells. In our studies, we did not explore any enzyme specific inhibition by pterostilbene. We would like to thank for the suggestion to perform such experiments. We have plans to conduct more detailed studies on the anticancer activity of pterostilbene against various melanoma cells and will present such results in our new article.
- Please mention standard used for the study.
Response:
To compare the interaction of pterostilbene with a chemotherapeutic drug with a proven strong cytostatic effect on cancer cells, doxorubicin was used at concentration 0,25 and
1 mM was used as a positive control in the study. The obtained results showed that doxorubicin at a concentration of 0.25 μM causes a very strong inhibition of the growth of melanoma cells. In cultures of the C32 line, it inhibited cell growth by 79% relative to the control (Fig. 2). A comparable or greater inhibitory effect on amelanotic cell growth was observed after incubation with pterostilbene at concentrations ≥ 20 μM. A2058 cultures treated with doxorubicin showed 88% growth inhibition in relation to control ( (Fig. 2). Pterostilbene at all concentrations tested caused a weaker inhibition of melanotic cell growth than doxorubicin.
|
A |
|
B |
|
Fig. 2 Cytotoxic effect of doxorubicin by the SRB method on (A) C32 and (B) A2058 melanoma cells.
In studies performed with the use of commercially available kits, the standards contained in the kits were used.
- Authors are informed to upload original images in form of supporting information for further clarifications.
Response:
In our study, we use spectrophotometric, RT-qPCR, and ELISA methods. We did not perform methods based on methods based on imaging (we used the ELISA method as an alternative to immunoblotting). Therefore, we cannot upload images. The original flow cytometry histograms we presented in the manuscript body.
- To study effect on p21 by pterostilbene that cause apoptosis can also be investigated using an Annexin V-PE/7-AAD Apoptosis Detection Kit and MTT assay. These assays can further support the confirmation of submitted research work.
Reference:
Guo Q, Wang D, Liu Z, Li C. Effects of p21 Gene Down-Regulation through RNAi on Antler Stem Cells In Vitro. PLoS One. 2015 Aug 26;10(8):e0134268. doi: 10.1371/journal.pone.0134268. PMID: 26308075; PMCID: PMC4550451.
Response:
We would like to thank the Reviewer for helpful comments. In the presented studies we decided to study the occurrence of irreversible stages of apoptosis in the by measuring caspase 3 activity and the formation of oligonucleosides. Before these assays we also performed XTT assay which is very similar to MTT assay. The XTT method detects cellular metabolic activities and, like the MTT assay, it is also based on tetrazolium salts. The results obtained after 48 hours of treatment with pterostilbene also confirmed that C32 cells were more sensitive to pterostilbene treatment than A2058 cells (Fig.3).
|
A |
|
B |
|
Fig. 3 Evaluation of the influence of pterostilbene on mitochondrial dehydrogenase activity by the XTT method on (A) C32 and (B) A2058 melanoma cells after 48 hours of treatment
(the means ± SD; * p < 0.05 vs. control).
- The effect of pterostilbene on the expressions of caspase-3, caspase-9, Bcl-2 and Bax can be studied using western blot for further confirmation.
Reference:
Zhao Y, Jing Z, Lv J, Zhang Z, Lin J, Cao X, Zhao Z, Liu P, Mao W. Berberine activates caspase-9/cytochrome c-mediated apoptosis to suppress triple-negative breast cancer cells in vitro and in vivo. Biomed Pharmacother. 2017 Nov;95:18-24. doi: 10.1016/j.biopha.2017.08.045.
Please upload original images in form of supporting information for further clarifications
Response:
In our studies, we used the ELISA method as an alternative to immunoblotting. Both are based on immunodetection. We have chosen ELISA method as it allows to determine specific protein fully quantitatively in cell lysates containing a mix of various proteins. ELISA is more sensitive and immunoblotting is a more specific assay, and thus the choice depends on the purpose. Western Blot (WB) is to some degree also used to quantify, but is not nearly as reliable as ELISA for that purpose, while in ELISA you will not see any size changes to your protein as long as the epitope of the protein that is recognized in the ELISA is there. We have chosen the ELISA method as it allows to determine specific protein fully quantitatively in cell lysates containing a mix of various proteins. Moreover, the data presented by Oh et al. [1] indicated that the ELISA had a smaller data distribution and was more repeatable, compared to Western blot data. In our experiments, we did not perform the Western blot method (our laboratory does not have the equipment necessary to perform this analysis).
[1] Oh SH, Choi YB, Kim JH, Weihl CC, Ju JS. Comparisons of ELISA and Western blot assays for the detection of autophagy flux. Data Brief. 2017;13:696-699. doi: 10.1016/j.dib.2017.06.045
Once again, we thank the Reviewer for carefully reading our manuscript and providing all comments that helped us improve our work.

Reviewer 2 Report
Major concerns are as follows:
- It is not clear why only one melanotic and one amelanotic cell lines were tested in this study. At least 3 various cell lines for melanotic and 3 for amelanotic should be examined. Additionally, there is no information about the authors’ preference (choice) about these cell lines. Briefly, explain why C32 and A2058 cell lines were selected for this study?
- Figure 1. Provide information about the IC50 values for pterostilbene (PTB) in C32 and A2058 cell lines in the BrdU assay.
- Page 5. Figure 3. Please provide the F-statistics values and the corresponding degrees of freedom and p values from one-way ANOVA for both cell lines C32 and A2058.
- Page 5. Figure 3. How to explain an 8-fold increase in CDKN1A mRNA expression after PTB treatment (40uM) in C32 cells? The other concentrations (20 and 60 uM) increased mRNA expression only 3-times.
- Page 6. Figure 4. How to explain a 10-fold higher concentrations of p21 in C32 cells for the control group (PTB 0 uM) than in A2058 cells?
- Figure 6 on page 8. How to explain that the basic content of Caspase-3 activity in A2058 cells was 10-times higher than that in C32 cells after 48h?
- Conclusions on page 15. No generalization is allowed since only one melanotic and one amelanotic cell lines were tested in this study. Please, reformulate the conclusions derived directly from the results from this study.
Minor concerns:
- Legend to figure 2. Line 103 – In the legend to figure 2 there is a symbol “$” for (G2/M), while on the graph there is a symbol “&”. Which one is correct?
- Page 5, line 127. Number of Figure. It should be Figure 3 instead of Figure 4. Please correct the number.
- Page 7, Figure 5. The title of the right column: Please correct the number of the cell line. It should be A2058.
- Please, correct grammar and syntax. Line 377 – it should be “still” instead of “steel”. Line 426 – it should be "spectrophotometric".
Author Response
Response to the Reviewer’s Comments
The authors thanks the Reviewer #2 for his (her) detailed revision and helpful suggestions comments. We completed and corrected the manuscript according to the Reviewer’s suggestions.
Comments and Suggestions for Authors
Major concerns are as follows:
- It is not clear why only one melanotic and one amelanotic cell lines were tested in this study. At least 3 various cell lines for melanotic and 3 for amelanotic should be examined. Additionally, there is no information about the authors’ preference (choice) about these cell lines. Briefly, explain why C32 and A2058 cell lines were selected for this study?
Response:
We agree with Reviewer comment that to confirm the effect of the compound on amelanotic and melanotic cells, at least three different cell lines (of each type) should be used. Despite many reports on the anticancer properties of pterostilbene against various cancers, data on its activity against skin cancer, especially melanoma, are still limited. Therefore, we decided to evaluate the effect of pterostilbene on melanoma cells. We have chosen two melanoma lines highly tumorigenic in nude mice, which differ in their ability to melanin synthesis and possess the BRAF mutation. The difference in melanin content is important due to the occurrence of amelanotic melanoma, an aggressive and difficult to diagnose subtype of melanoma. We have chosen cells with BRAF mutation because approximately 50% of patients with melanoma have mutations in BRAF proto-oncogene. Activating mutations in BRAF kinase favor growth, differentiation, proliferation, migration, and apoptosis of melanoma cells. At present we plan to conduct research using a wide panel of melanoma cells that differ not only in melanin content but also in selected mutations. It seems especially important due to the high heterogeneity of melanoma.
- Response:
Figure 1. Provide information about the IC50 values for pterostilbene (PTB) in C32 and A2058 cell lines in the BrdU assay.
Response:
Thank you very much for the suggestion. We introduced into the manuscript IC50 values in Results section as well as complemented Materials and Methods part.
- Page 5. Figure 3. Please provide the F-statistics values and the corresponding degrees of freedom and p values from one-way ANOVA for both cell lines C32 and A2058.
Response:
The F-statistics, degrees of freedom an p values from one-way ANOVA we presented in table below.
|
Gene |
Cell line |
Analysis of Variance |
||
|
|
df |
F |
p |
|
|
CCND1 |
C32 |
3 |
12,6668 |
0,0020 |
|
CCND1 |
A2058 |
3 |
6,3089 |
0,0167 |
|
CDKN1A |
C32 |
3 |
31,1737 |
<0,001 |
|
CDKN1A |
A2058 |
3 |
7,0108 |
0,0125 |
df-degrees of freedom, F-statistics, p- statistically significant if p < 0,05
- Page 5. Figure 3. How to explain an 8-fold increase in CDKN1A mRNA expression after PTB treatment (40uM) in C32 cells? The other concentrations (20 and 60 uM) increased mRNA expression only 3-times.
Response:
We also noticed that difference between expression of CDKN1A gene in cells incubated with 40 mM pterostilbene and cells incubated with other concentrations of pterostilbene. Unfortunately we cannot explain that results. However we did not studied the expression of CDKN1A gene at other time points. It is possible that the highest concentration of pterostilbene can cause a stronger increase in transcriptional activity of this gene at earlier time points and the lowest concentration after longer incubation. It also is worth mentioning that that incubation of C32 cells with pterostilbene at concentration 40 mM caused more pronounced effect on the expression of CASP3, CASP9, BAX genes, caspase 3 activity, and DNA fragmentation after 48 h than pterostilbene at concentrations 20 mM and 40 mM.
- Page 6. Figure 4. How to explain a 10-fold higher concentrations of p21 in C32 cells for the control group (PTB 0 uM) than in A2058 cells?
Response:
P21 overexpression could be the result of cumulative gene mutations. In melanomas, higher p21 expression was associated with increasing Breslow thickness. The expression of the p21 protein has been shown to be lower in thinner melanomas and increases in more advanced melanomas [1]. The basal level of the p21 protein could also be important in light of newly published research on the relationship between p21 expression and sensitivity of melanoma cells to targeted therapies [2].
The role of the p21 protein in cancer cells is complexed. It exerts inhibitory effects on p53 and apoptosis. Furthermore, p21 can also possess the anti-apoptotic property that relies on its ability to inhibit the activity of proteins directly involved in the induction of apoptosis, including the caspase cascade and stress-activated protein kinases, as well as down-regulation of pro-apoptotic genes and up-regulation of factors with anti-apoptotic activities. Therefore, the pivotal role of p21 in the determination of cell fate is also through the control of gene expression and apoptosis by acting at different levels of the death cascade [3]. Therefore, there is the possibility that the pterostilbene-induced increase in the level of the p21 protein in A2058 cells could be related to its resistance to caspase-dependent apoptosis.
[1] de Sá BC, Fugimori ML, Ribeiro Kde C, Duprat Neto JP, Neves RI, Landman G. Proteins involved in pRb and p53 pathways are differentially expressed in thin and thick superficial spreading melanomas. Melanoma Res. 2009 Jun;19(3):135-41. doi: 10.1097/CMR.0b013e32831993f3. PMID: 19369901.
[2] Fröhlich LM, Makino E, Sinnberg T, Schittek B. Enhanced expression of p21 promotes sensitivity of melanoma cells towards targeted therapies. Exp Dermatol. 2022 Aug;31(8):1243-1252. doi: 10.1111/exd.14585. Epub 2022 Jun 2. PMID: 35514255.
[3] Mirzayans R, Andrais B, Kumar P, Murray D. Significance of Wild-Type p53 Signaling in Suppressing Apoptosis in Response to Chemical Genotoxic Agents: Impact on Chemotherapy Outcome. International Journal of Molecular Sciences. 2017; 18(5):928. https://doi.org/10.3390/ijms18050928
- Figure 6 on page 8. How to explain that the basic content of Caspase-3 activity in A2058 cells was 10-times higher than that in C32 cells after 48h?
Response:
Unfortunately we cannot exactly explain the reason of high basic Caspase-3 activity in A2058 cells. Studies carried out by Madrit et al. [1] revealed that the basal level of Caspase-3 activity in A2058 cells after 48 h was 0.27 ng/mg protein. These results are in line with the findings of the current studies (activity after 48 h was 290 pg/mg protein = 0,29 ng/mg protein).We suppose that the high base level of Caspase-3 could be the specificity of these cells. Many tumor cell lines and tissues has been reported to have high basal Caspase-3 activity what could be caused by the molecular changes associated with malignant transformation. The current literature contains several examples of new biological functions of caspases other than apoptosis. For example, there are involved in cell differentiation and may activate T cells that could further influence tumor biology through inflammatory factors. Basal caspase-3 activity has been reported to induce glioblastoma cell migration without apoptotic stimulation. In melanoma, basal caspase 3 expression seems to correlate with invasion potential [2,3].
[1] Madrid A, Cardile V, González C, Montenegro I, Villena J, Caggia S, Graziano A, Russo A. Psoralea glandulosa as a potential source of anticancer agents for melanoma treatment. Int J Mol Sci. 2015 Apr 9;16(4):7944-59. doi: 10.3390/ijms16047944. PMID: 25860949; PMCID: PMC4425060.
[2] Donato AL, Huang Q, Liu X, Li F, Zimmerman MA, Li CY. Caspase 3 promotes surviving melanoma tumor cell growth after cytotoxic therapy. J Invest Dermatol. 2014 Jun;134(6):1686-1692. doi: 10.1038/jid.2014.18. Epub 2014 Jan 16. PMID: 24434746; PMCID: PMC4020991.
[3] Liu YR, Sun B, Zhao XL, Gu Q, Liu ZY, Dong XY, Che N, Mo J. Basal caspase-3 activity promotes migration, invasion, and vasculogenic mimicry formation of melanoma cells. Melanoma Res. 2013 Aug;23(4):243-53. doi: 10.1097/CMR.0b013e3283625498. PMID: 23695439.
- Conclusions on page 15. No generalization is allowed since only one melanotic and one amelanotic cell lines were tested in this study. Please, reformulate the conclusions derived directly from the results from this study.
Response:
We agree with the Reviewer that generalization is not allowed so that we reformulated Conclusion part.
Minor concerns:
- Legend to figure 2. Line 103 – In the legend to figure 2 there is a symbol “$” for (G2/M), while on the graph there is a symbol “&”. Which one is correct?
Response:
We apologize for the mistake. We corrected the legend to figure 2.
- Page 5, line 127. Number of Figure. It should be Figure 3 instead of Figure 4. Please correct the number.
Response:
The number of Figure on the page 5 has been corrected.
- Page 7, Figure 5. The title of the right column: Please correct the number of the cell line. It should be A2058.
Response:
The number of cell line in Figure 5 has been corrected.
- Please, correct grammar and syntax. Line 377 – it should be “still” instead of “steel”. Line 426 – it should be "spectrophotometric".
Response:
We checked the manuscript and corrected the grammar and syntax errors.

Reviewer 3 Report
Wawszczyk et al evaluate the influence of pterostilbene on the proliferation and apoptosis of human melanoma cells, using BRDU assay, flow cytometry, and real-time RT-qPCR and ELISA. They found that pterostilbene reduced the proliferation of melanoma cells and that pterostilbene induced apoptosis in amelanotic C32 melanoma cells mediated by an increase in the expression of genes encoding BAX, caspase-9, and caspase 3, induction of caspase 3 activity, and DNA degradation. They also found pterostilbene did not affect the activation of apoptosis in the A2058 cell line. Overall, this study is very clear and no big caveat were observed. However, a major revision is still needed before acceptance, as the reviewer still have the following points of concern that need to be addressed.
1. This is my biggest concern. The authors only tested this drug on two cell lines, one melanotic and the other amelanotic. This is not enough, and many findings and claims from the authors can be wrongly arrived at due to the variability of only two cell lines. Similar assays need to be performed in a large panel of cell lines.
2. It was not clear why pterostilbene would work only in one but not the other cell line. This is important to know. Deep molecular profiling (such as RNA-seq) will be needed to better infer the mechanisms of differential sensitivity across cell lines.
3. The fact that the melanocytic level of cell lines has differential sensitivity to pterostilbene is very interesting. It has already been discovered (PMID: 29229836, 28128880, 29657129, ) in the context of melanoma for BRAF inhibitors that the more melanocytic phenotypes are more sensitive to BRAF inhibitors and the less melanocytic more mesenchymal ones are less sensitive. Could this be the same case here for pterostilbene? This will be important to test and discuss within the context of these published literature.
4. Metabolic differences and their association with re-sensitize the pterostilbene non-responding cell line. The authors observed the differences in the sensitivities to pterostilbene across two cell lines with different melanocytic levels. Will it be possible that there are some metabolic differences between the two cell lines which can be utilized to re-sensitive the pterostilbene non-responding cell line? It has been reported (PMID: 32973134) that the melanocytic cell lines are more sensitive to fatty acid synthesis inhibitor and the non-melanocytic ones are more sensitive to lipid mono-unsaturation. These drugs will be worthwhile to test and discuss within the context of the literature.
5. More discussion on the heterogeneity of melanoma cell lines and its association with drug resistance? The discussion section of the current manuscript can be further improved. One of the hot topics in cancer is tumor heterogeneity and its association with drug response and resistance. In melanoma, it has already been found (PMID: 32393797, 28128880, 29229836, 31166947) that some melanoma cell lines are a mixture of different subpopulations and different subpopulations within the same cell line may have differential sensitivities to a drug. This will be an important aspect to be included in the discussion to further elevate the novelty of the paper.
Author Response
Response to the Reviewer’s Comments
The authors thanks the Reviewer #3 for his (her) detailed revision and valuable comments. We completed and corrected the manuscript according to the Reviewer’s suggestions.
Comments and Suggestions for Authors
Wawszczyk et al evaluate the influence of pterostilbene on the proliferation and apoptosis of human melanoma cells, using BRDU assay, flow cytometry, and real-time RT-qPCR and ELISA. They found that pterostilbene reduced the proliferation of melanoma cells and that pterostilbene induced apoptosis in amelanotic C32 melanoma cells mediated by an increase in the expression of genes encoding BAX, caspase-9, and caspase 3, induction of caspase 3 activity, and DNA degradation. They also found pterostilbene did not affect the activation of apoptosis in the A2058 cell line. Overall, this study is very clear and no big caveat were observed. However, a major revision is still needed before acceptance, as the reviewer still have the following points of concern that need to be addressed.
- This is my biggest concern. The authors only tested this drug on two cell lines, one melanotic and the other amelanotic. This is not enough, and many findings and claims from the authors can be wrongly arrived at due to the variability of only two cell lines. Similar assays need to be performed in a large panel of cell lines.
Response:
We agree with Reviewer comment that to confirm the effect of the compound on amelanotic and melanotic cell, more cell lines should be used. Despite many reports on the anticancer properties of pterostilbene against various cancers, data on its activity against skin cancer, especially melanoma, are still limited. Therefore, we decided to conduct a preliminary study to determine whether pterostilbene affects melanoma cell lines. We have chosen two melanoma lines highly tumorigenic in nude mice, which differ in their ability to melanin synthesis and possess the same type of BRAF mutation (activation of BRAF via a V600E). The difference in melanin content is important due to the occurrence of amelanotic melanoma, an aggressive and difficult to diagnose subtype of melanoma. We have chosen cells with the BRAF mutation because approximately 50% of patients with melanoma have mutations in the BRAF proto-oncogene. Activating mutations in BRAF kinase favor growth, differentiation, proliferation, migration, and apoptosis of melanoma cells. At present we plan to conduct a more detailed study with the use a wide panel of melanoma cells that differ not only in melanin content but also in selected mutations. It seems especially important due to the high heterogeneity of melanoma.
- It was not clear why pterostilbene would work only in one but not the other cell line. This is important to know. Deep molecular profiling (such as RNA-seq) will be needed to better infer the mechanisms of differential sensitivity across cell lines.
Response:
We agree with Reviewer that to confirm the exact molecular reason of different response studied melanoma cell to pterostilbene treatment more detailed studies are needed. We would like to thank for the suggestion to use molecular profiling. We have the plans to perform such deeper studies with the use a wide panel of melanoma.
- The fact that the melanocytic level of cell lines has differential sensitivity to pterostilbene is very interesting. It has already been discovered (PMID: 29229836, 28128880, 29657129, ) in the context of melanoma for BRAF inhibitors that the more melanocytic phenotypes are more sensitive to BRAF inhibitors and the less melanocytic more mesenchymal ones are less sensitive. Could this be the same case here for pterostilbene? This will be important to test and discuss within the context of these published literature.
Response:
As suggested by the Reviewer, we discussed the relationship between the melanocytic level of and differential sensitivity to treatment in the Discussion section, as well as cited the recommended interesting papers.
- Metabolic differences and their association with re-sensitize the pterostilbene non-responding cell line. The authors observed the differences in the sensitivities to pterostilbene across two cell lines with different melanocytic levels. Will it be possible that there are some metabolic differences between the two cell lines which can be utilized to re-sensitive the pterostilbene non-responding cell line? It has been reported (PMID: 32973134) that the melanocytic cell lines are more sensitive to fatty acid synthesis inhibitor and the non-melanocytic ones are more sensitive to lipid mono-unsaturation. These drugs will be worthwhile to test and discuss within the context of the literature.
Response:
We would like to thank Reviewer for the suggestion. According to the Reviewer’s recommendation, we introduced the mentioned metabolic difference of melanoma cells into the Discussion. Obviously, the list of references has been updated.
- More discussion on the heterogeneity of melanoma cell lines and its association with drug resistance? The discussion section of the current manuscript can be further improved. One of the hot topics in cancer is tumor heterogeneity and its association with drug response and resistance. In melanoma, it has already been found (PMID: 32393797, 28128880, 29229836, 31166947) that some melanoma cell lines are a mixture of different subpopulations and different subpopulations within the same cell line may have differential sensitivities to a drug. This will be an important aspect to be included in the discussion to further elevate the novelty of the paper.
Response:
We discuss the heterogeneity in melanomas and its influence on difficulties in the effective treatment of melanoma. Thank you for suggestions that helped us improve our work.
Once again, we thank the reviewer for careful reading of our manuscript and providing all comments and suggestions.

Round 2
Reviewer 1 Report
Manuscript " Pterostilbene-mediated inhibition of cell proliferation and cell death induction in amelanotic and melanotic melanoma" is now recommended for publication.
Author Response
Response to the Reviewer’s Comments
We thank Reviewer for his (her) for the reviewing of our manuscript. We appreciate the positive feedback from the Reviewer.

Reviewer 2 Report
No further comments.
Author Response

(The authors gave the same response as above.)

Reviewer 3 Report
The revised version of the manuscript has addressed some of my concerns and has been improved when compared with the initial draft.
However, I still think the discussion regarding the heterogeneity of melanoma and its association with response to therapy can be further improved.
For example, it has already been shown (PMID: 32393797) that for a clonal melanoma cell line, there can be the co-existence of two different subpopulations which may lead to two different trajectories toward the tolerance of a drug. Thus, using combination therapy to simultaneously co-targeting these different paths towards drug tolerance is needed to fully prevent drug tolerance. Further, the trajectories towards drug tolerance are driven by the competing forces between phenotype stability and phenotype-proliferation-rates (PMID: 31166947).
It will be beneficial to discuss these examples to further illustrate the importance of heterogeneity and "how" to use such heterogeneity information to design better treatment.
Author Response
Response to the Reviewer’s Comments
The authors thanks the Reviewer #3 for his (her) valuable comments. We completed and corrected the manuscript according to the Reviewer’s suggestions.
Comments and Suggestions for Authors
The revised version of the manuscript has addressed some of my concerns and has been improved when compared with the initial draft.
However, I still think the discussion regarding the heterogeneity of melanoma and its association with response to therapy can be further improved.
For example, it has already been shown (PMID: 32393797) that for a clonal melanoma cell line, there can be the co-existence of two different subpopulations which may lead to two different trajectories toward the tolerance of a drug. Thus, using combination therapy to simultaneously co-targeting these different paths towards drug tolerance is needed to fully prevent drug tolerance. Further, the trajectories towards drug tolerance are driven by the competing forces between phenotype stability and phenotype-proliferation-rates (PMID: 31166947).
It will be beneficial to discuss these examples to further illustrate the importance of heterogeneity and "how" to use such heterogeneity information to design better treatment.
Response:
We would like to thank the Reviewer for the suggestion that helped us improve our work. The Discussion section in the revised version of the manuscript was modified according to the suggestions of the reviewer (line 442-487). Obviously, the list of references has been updated.
Once again, we thank the reviewer for careful reading of our manuscript and providing all comments and suggestions.
